

# A systematic metadata harvesting workflow for analysing scientific networks

Bilal H. Butt[1], Muhammad Rafi[2] and Muhammad Sabih[3]

[1] Department of Computer Science, D.H.A. Suffa University, Karachi, Pakistan
[2] Department of Computer Science, National University of Computer and Emerging Sciences, Karachi, Pakistan
[3] Department of Electrical Engineering, D.H.A. Suffa University, Karachi, Pakistan

## ABSTRACT

One of the disciplines behind the science of science is the study of scientific networks. This work focuses on scientific networks as a social network having different nodes and connections. Nodes can be represented by authors, articles or journals while connections by citation, co-citation or co-authorship. One of the challenges in creating scientific networks is the lack of publicly available comprehensive data set. It limits the variety of analyses on the same set of nodes of different scientific networks. To supplement such analyses we have worked on publicly available citation metadata from Crossref and OpenCitatons. Using this data a workflow is developed to create scientific networks. Analysis of these networks gives insights into academic research and scholarship. Different techniques of social network analysis have been applied in the literature to study these networks. It includes centrality analysis, community detection, and clustering coefficient. We have used metadata of Scientometrics journal, as a case study, to present our workflow. We did a sample run of the proposed workflow to identify prominent authors using centrality analysis. This work is not a bibliometric study of any field rather it presents replicable Python scripts to perform network analysis. With an increase in the popularity of open access and open metadata, we hypothesise that this workflow shall provide an avenue for understanding scientific scholarship in multiple dimensions.

## INTRODUCTION

Scientific networks provide useful information in understanding the dynamics of science (*Price, 1965*). With the advent of numerous bibliographic data sources (*Waltman & Larivière, 2020*), it is now possible to analyse different scientific networks. The proposed study focuses on article citation network, author citation network, and co-authorship network. Usually, studies that focus on co-authorship do not require information about the citation. However, having citation links would enable a more complete and holistic view of the possible relations among authors (*Zingg, Nanumyan & Schweitzer, 2020*). To achieve this, objective comprehensive access to citation metadata is required. This can be accomplished using publicly available citation metadata available via Crossref (*Hendricks et al., 2020*). However, applying network analysis on it requires a series of steps

Corresponding author
Bilal H. Butt, bilal.hayat@dsu.edu.pk

that may not be intuitive. The proposed study furnishes details of these steps so that it is easy to supplement it with different analyses.

Social network analysis techniques are applied to study scientific networks. It includes citation networks of article or author, and author collaboration network. Usually, these networks are build using different data sources. However, our workflow can create all these networks using OpenCitations data and Crossref. The workflow presented in this article is part of a study on the influence of scholarly research artefacts. To this end, we primarily limit our research goal to have a systematic workflow for analysing scientific networks. In this work, we aim to utilise open metadata (*Peroni et al., 2015*), made available using Crossref. Also, we utilise open source Python libraries for network analysis, namely, NetworkX (*Hagberg, Schult & Swart, 2008*) and SNAP (*Leskovec & Sosič, 2016*). Python is used based on its popularity with researchers as per survey results by *AlNoamany & Borghi (2018)*. Although graphical software has an ease of use, we prefer to provide workflow as a set of Python scripts to facilitate advance analysis. Details of batch execution of workflow scripts are available on GitHub for researchers with programming background (*Butt & Faizi, 2020*). This article outlines details of a case study for analysing collaboration network of Scientometrics journal metadata. All steps are documented for the replication of this study. This work shall lay the groundwork for analysing scientific networks using metadata of different journals, set of journals or a subject category. One such analysis is the identification of prominent authors (gurus).

Identifying prominent authors of any field is one of the primary focus for young researchers. Likewise, other researchers tend to follow research published by gurus of the field. Defining a guru of the field is not an easy task, and the definition of guru will be very subjective. To this end, we focus on the definition of guru using the centrality measures of social network analysis. Details of different centrality measures are depicted in Fig. 1 (*Newman, 2010*). The following descriptions were inspired by *Milojević (2014)*. Simply put, any author with a high citation count may be considered the guru. It can be achieved using degree centrality. Another way of identifying a highly cited individual is to calculate whose article is in top percentile within the domain. However, we currently limit such definitions to degree centrality of articles. It is not always the case that all highly cited authors are equally influential. Those who are cited by other influential authors may also be termed as influential even though they may or may not have high citation count. Likewise, any author frequently collaborating with influential authors would also influence that field. This recursive definition of influence is well captured by eigenvector centrality. Another centrality measure, namely betweenness centrality would define an author as prominent if author collaborates with different groups. Centrality measures of closeness and farness is the extent to which an author is on average close to or far from other authors within the network, respectively.

In the case of analysing the citation network with a limited snapshot of data, this could be supplemented by creating the ego-centered network (*Newman, 2003*). Citation index allow fetching the metadata of articles in two directions. First, articles cited by the original article can be accessed as references. Second, articles that cites the original article can be accessed as citations. For each node, its connecting nodes inclusion enhance

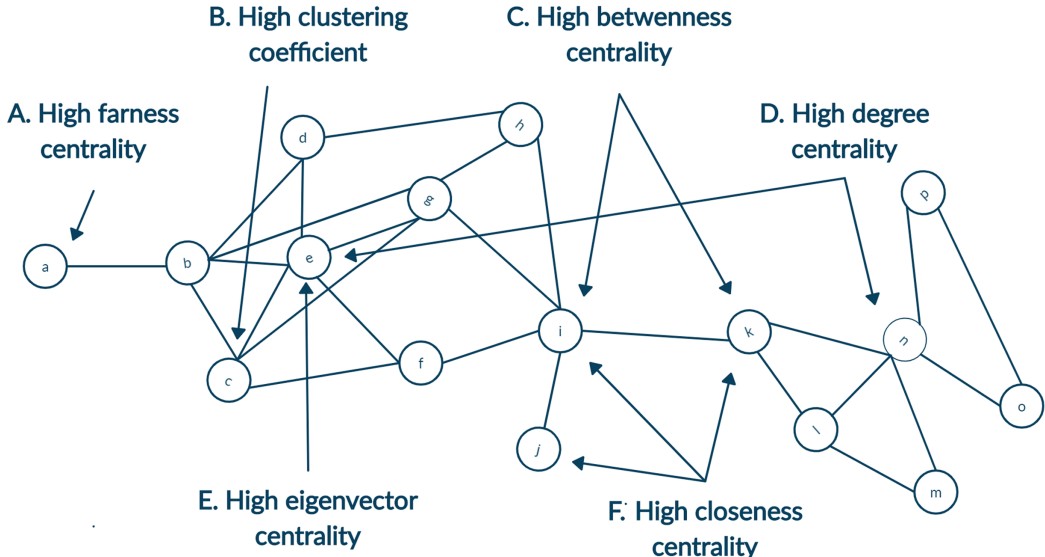

**Figure 1** **A toy network showing nodes with high centrality for different measures.** (A) High farness centrality since the node a has the maximum average distance to other nodes. (B) High clustering coefficient since neighbours of the node c are all connected as well. (C) High betweenness centrality since the highest number of shortest paths will go from the node i and k since they are bridging two parts of the network. (D) High degree centrality as both the nodes e and n have highest count of neighbours. (E) High eigenvector centrality since node e is connected to many neighbours with a relatively higher degree. (F) High closeness centrality as the average distance from nodes i, j and k are minimum to other nodes.

the breadth of the network. Node is known as the ego and its neighbours are termed as alters. In Fig. 1 alters of ego node n, namely, node k, l, m, o and p will form its ego network.

The remainder of this article is structured as follows. After giving some preliminary definitions, an overview of research is presented on scientific networks using centrality analysis. Next, we provide details about the pipeline architecture in methodology before moving to the steps of acquiring and analysing data. Steps include extracting details of the citation index which is downloaded from the web and loaded in memory. Next, we discuss how the citation metadata are fetched and filtered. In the last step, we explain the analysis of different scientific networks. Further, a case study is discussed for applying centrality analysis on collaboration network. We conclude with details of how this work can be further enhanced.

## PRELIMINARIES

Some terminologies are introduced below that are necessary for understanding this study. Also, some acronyms used in this article are listed.

**SNA** Social network analysis is concerned with techniques and measures applied to real-world networks

**Centrality** A property of the node which shows the relative rank of a node compared to other nodes

**Degree centrality** Nodes having a high number of connections are more central

**Betweenness centrality** Nodes on the majority of the shortest path connecting other pair of nodes

**Closeness centrality** Average distance to other nodes in the network is minimum

**Farness centrality** Reciprocal of closeness centrality

**Clustering coefficient** Well knitted alliances get a higher score

**Eigenvector centrality** Influence of one node transfers to other connected nodes

**PageRank** Variant of eigenvector centrality along with provision for randomness

**Katz centrality** Variant of eigenvector centrality for directed graphs without cycles

**Guru** Any important node in the scientific network as defined by centrality measures

**Scientific network** A social network with nodes as author/article and edges as co-author/citation

**Citation network** A type of scientific network with article nodes connected by cited-to and citing edge

**Collaboration network** A type of scientific network with author nodes connected by co-author edge

**Co-citation network** A type of scientific network with nodes connected if both nodes get cited together

**Bibliographic coupling** A type of scientific network with nodes connected if both cites the same node

**Ego network** Include all nodes connected to and by any specific (ego) node

**JSON** JavaScript Object Notation is s format to represent complex and unstructured data

**Crossref** Using api.crossref.org for publicly available citation data (*Hendricks et al., 2020*)

**COCI** The OpenCitations Index of Crossref open DOI-to-DOI citations (*Heibi, Peroni & Shotton, 2019b*)

**WoS** Clarivate Analytics' Web of Science www.webofknowledge.com (*Birkle et al., 2020*)

**JCR** Journal Citation Reports incites.clarivate.com

**Scopus** Elsevier's Scopus www.elsevier.com/solutions/scopus (*Baas et al., 2020*)

**DBLP** Digital Bibliography and Library Project www.dblp.org

**ACM** ACM Digital Library dl.acm.org

**APS** American Physical Society www.aps.org

**MAG** Microsoft Academic Graph academic.microsoft.com (*Wang et al., 2020*)

**PubMed** PubMed Central Database www.ncbi.nlm.nih.gov/pmc

**CiteSeerX** Scientific literature digital library and search engine citeseerx.ist.psu.edu

**ACL** ACL Anthology Reference Corpus acl-arc.comp.nus.edu.sg

**SNAP** Stanford Network Analysis Project snap.stanford.edu

**SCIM** Scientometrics Journal www.springer.com/journal/11192

**API** Application Programming Interface
**Table 1 Review of studies applying social network analysis on scientific networks.**

| Study | Bibliographic data source | Scientific network(s) | Social network analysis |
|---|---|---|---|
| *Ding (2011)* | WoS | Author Citation | Weighted PageRank |
| *Abbasi, Hossain & Leydesdorff (2012)* | Scopus | Author Collaboration | Degree, Betweenness, Closeness Centrality |
| *Ortega (2014)* | MAG | Co-Author Ego Network | Clustering Coefficient, Degree and Betweenness Centrality |
| *Milojević (2014)* | WoS | Author Collaboration and Citation, Article Citation | Degree, Betweenness, Closeness, Eigenvector Centrality |
| *Waltman & Yan (2014)* | WoS | Journal Citation Network | PageRank |
| *Xu & Pekelis (2015)* | Manual | Author Citation | PageRank and Degree Centrality |
| *Leydesdorff, Wagner & Bornmann (2018)* | WoS/JCR | Journal Citation | Betweenness Centrality |
| *Lee (2019)* | Scopus | Author Collaboration | Degree and Betweenness Centrality, Clustering Coefficient |
| *Massucci & Docampo (2019)* | WoS | Institutional Citation | PageRank |
| *Singh & Jolad (2019)* | APS Journals | Author Collaboration | Centrality, Community Detection |
| *Van den Besselaar & Sandström (2019)* | Manual | Researchers Ego Network | Clustering coefficient, Eigenvector Centrality |
| *Waheed et al. (2019)* | DBLP, ACM, MAG | Author Collaboration and Citation, Article Citation, Co-citation and Bibliographic Coupling | Degree, Betweenness, Closeness, Eigenvector Centrality |

# RELATED WORK

Visualising bibliographic data as a network is not new, *Price (1965)* introduced the work more than 50 years ago. Most recent studies are on co-authorship network (*Abbasi, Hossain & Leydesdorff, 2012*; *Milojević, 2014*; *Lee, 2019*; *Singh & Jolad, 2019*; *Waheed et al., 2019*), however others have focused on citation network of authors (*Ding, 2011*; *Milojević, 2014*; *Xu & Pekelis, 2015*; *Waheed et al., 2019*) or citation network of journals (*Waltman & Yan, 2014*; *Leydesdorff, Wagner & Bornmann, 2018*). Only a couple of studies have utilised more than one scientific network for analysis (*Milojević, 2014*; *Waheed et al., 2019*). Traditionally bibliometric analysis has been done using WoS and Scopus (*Waltman & Larivière, 2020*). A similar case has been observed in studies on scientific network analysis where the data sources used are Scopus (*Abbasi, Hossain & Leydesdorff, 2012*; *Lee, 2019*) or WoS (*Ding, 2011*; *Milojević, 2014*; *Waltman & Yan, 2014*; *Leydesdorff, Wagner & Bornmann, 2018*; *Massucci & Docampo, 2019*). However, some recent studies have focused on open access data sources (*Singh & Jolad, 2019*; *Van den Besselaar & Sandström, 2019*; *Waheed et al., 2019*). Other data sources such as PubMed, CiteSeerX and ACL are not discussed in this article. They are used mostly for text analysis instead of network analysis. Below we list a brief account of work done on scientific networks using centrality measures. Details are summarised in Table 1 in chronological order. Some earlier studies such as (*Newman, 2004*) are not included as we have focused on studies published in the last decade.

**Butt et al. (2021), *PeerJ Comput. Sci.*, DOI 10.7717/peerj-cs.421**                                                                  **5/19**

*Ding (2011)* proposed to analyse the author citation network with weighted PageRank. The author proposed the strategy on predicting prize winners that outperforms the conventional h-index and related citation count measures. *Abbasi, Hossain & Leydesdorff (2012)* discussed the use of betweenness centrality as a measure of getting more collaborators compared to degree and closeness centrality. They have used temporal co-authorship network in the steel research domain. Data was manually curated and downloaded from Scopus.

*Ortega (2014)* analysed 500 co-authors' ego network and conclude that bibliometric indicators and centrality measures are correlated. They have used clustering coefficient, degree and betweenness centrality as local metrics. Some global level metrics were also analysed using the ego network. It is one of the early studies using MAG.

Two book chapters provide hands-on details about centrality measures (*Milojević, 2014*) and PageRank (*Waltman & Yan, 2014*) using WoS data. *Milojević (2014)* constructed the author collaboration network and calculated degree, betweenness, eigenvector and closeness centrality. *Waltman & Yan (2014)* details applying PageRank on journal citation network.

*Xu & Pekelis (2015)* used a manually curated dataset for authors of China and Taiwan in the field of Chinese Language Interpreting Studies. They have applied PageRank and degree centrality to find influential authors within different clusters identified using community detection.

*Leydesdorff, Wagner & Bornmann (2018)* have used betweenness centrality to measure multidisciplinary journals. Authors have limited their approach with a diversity measure and evaluated it on JCR data. Usually, a journal gets citation within its subject category but those journals cited/citing the other fields are considered a bridge between the subject categories.

A case study for young researchers performance evaluation is presented by *Lee (2019)*. The author analysed the collaboration network of these researchers using statistical analysis for the frequency of collaborators. The degree centrality is showed to correspond with future publication count. It is akin to *Li et al. (2019)* who concludes that collaboration of young scientist with top-ranked co-authors has a high probability of future success.

*Massucci & Docampo (2019)* applied the PageRank algorithm on a university citation network. Working on five different subject categories, they proposed a framework which is more robust than existing university rankings. It holds a high correlation with these accepted rankings. *Singh & Jolad (2019)* utilised data of APS journals to form collaboration network of Indian physicist. In this co-authorship network, they have applied different centrality measures and report the overlapping top authors.

*Van den Besselaar & Sandström (2019)* discuss the potential use of clustering coefficient and eigenvector centrality in ego network of research students and their supervisor. Both metrics are used to gauge the independence of a researcher. They have handpicked four pairs of researchers. The authors suggested that there are numerous ways to capture the researcher's autonomy. However, when evaluating large data sets the clustering coefficient and eigenvector centrality can be effective.

*Waheed et al. (2019)* discusses the use of centrality measures on multiple scientific networks to improve article recommendation. They filter the citation network to five levels in cited-by and citing directions. Evaluating a large-scale network available at AMiner they proposed a hybrid recommendations strategy. It includes different centrality measures on author collaboration network, author citation network and article citation network.

Table 1 summarises the studies in three aspects. First, the bibliographic data source used. Second, the scientific network created. Last, details of techniques applied for analysis. Studies show that applying centrality measures is a useful analysis in bibliometrics. However, these approaches are mostly not scalable and require considerable effort to apply the same analysis on bigger networks. In some cases, the tools limit the size of network analysed, whereas in other studies the data sets are manually curated. In comparison to our work, most of the studies are limited to one type of network. The way data sets are acquired limits the analysis to expand to another type of networks (*Zingg, Nanumyan & Schweitzer, 2020*). We observe that very few studies have either used multiple networks or mentioned that if these can be curated with the same data source. With WoS and Scopus, it is theoretically possible to create all networks. However, with other data sources, a dump is usually uploaded with limited metadata. It restricts the authors to confine their studies.

Case studies similar to our workflow are also available on the proprietary data source of WoS (*Milojević, 2014*) and Scopus (*Rose & Kitchin, 2019*). Further, a set of graphical tools are also available as discussed by *Moral-Muñoz et al. (2020)* in a recent survey. Most tools do not give access for Crossref apart from *Van Eck & Waltman (2014)*, *Chen (2005)*.

One of the recent studies (*Rose & Kitchin, 2019*) focuses on using Scopus data for network analysis. They have provided an API for researchers to perform useful analyses. Accessing Scopus is possible with Elsevier Developer API Key. However, it requires institutional or authenticated access. Such access is not possible, especially for developing countries (*Herzog, Hook & Konkiel, 2020*). Although our work is similar to *Rose & Kitchin (2019)* in using Python for analysing scientific network, it is different in two aspects. Firstly, we are working with OpenCitatons data using Crossref. Secondly, we have not developed an API interface that needs maintenance and support since Crossref, NetworkX and SNAP fulfil the purpose.

*Chen (2005)* discusses the identification of highly cited clusters of a scientific network. The pivotal points in the scientific network are captured using betweenness centrality. The author uses clinical evidence data associated with reducing risks of heart disease to illustrate the approach. They have discussed the design of the CiteSpace tool and its new feature for identifying pivotal points. They used betweenness centrality to identify pathways between thematic clusters. Nodes with high betweenness centrality are potential pivotal points in clustering the scientific network. Instead of a graphical software tool, we propose to use Python scripts. It gives more flexibility for advance analysis. For a detailed survey, we would refer the interested reader to *Moral-Muñoz et al. (2020)*.

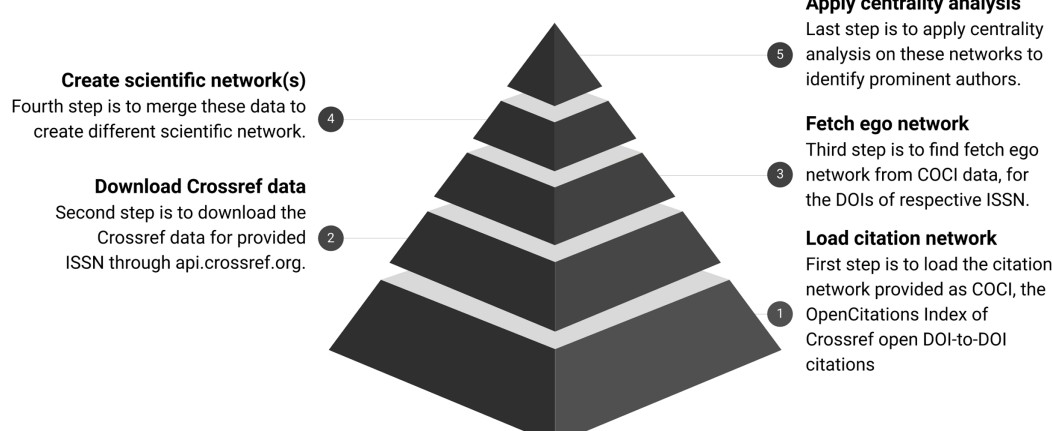

**Figure 2  Workflow for analysing scientific networks.** The pyramid shows the refinement of data at every step. COCI contains approximately 625 M edges. It gets reduced as a subset of nodes fetched for respective ISSN. Finally, the top of the pyramid shows the output in the form of a few nodes identified with high centrality.                               

Usage of open metadata are gaining popularity. On the flip side, publicly available metadata has its limitations with completeness and verification. *Iorio, Peroni & Poggi (2019)* concludes that using OpenCitatons data for evaluation purpose is not enough due to the unavailability of complete data. However, more than half of the records are available in comparison to WoS and Scopus. A similar evaluation is also done by *Nishioka & Färber (2019)* and *Martín-Martín et al. (2020)*. Further, there are different approaches to augment the current OpenCitatons data (*Daquino et al., 2018*; *Heibi, Peroni & Shotton, 2019a*; *Peroni & Shotton, 2020*). *Kamińska (2018)* discusses a case study for using OpenCitatons data for visualising citation network. *Zhu et al. (2020)* has used COCI to evaluate books scholarship. With a scripted workflow, we hypothesise that it would be easier for masses to adopt OpenCitatons data for bibliometric analysis.

## METHODOLOGY

This section details a systematic workflow from data fetching to analysis. A series of steps are required to apply centrality analysis on the author collaboration and author citation networks. Utilising the article citation network, available as citation index, these networks get created. All scripts were executed on Windows Server machine having Quad-Core AMD Opteron (TM) Processor 6272 with 128 GB RAM installed. The initial processing of data requires heavy computation and memory once. Later, the data are converted to a compressed binary format using libraries for processing large networks. It can run on any standard laptop machine. Below, we provide details of the workflow to create scientific networks. Although the case study is limited to data of SCIM, we have made the process automated. This automation helps applying the same script for other journals with minimum changes.

Overview of the process is shown in Fig. 2 and further details about each of the following steps are documented separately. Each step is further distributed with three sub-steps for clarity and batch execution.

**Step 1** Download the citation index available as COCI (*Heibi, Peroni & Shotton, 2019b*).
**Step 2** Download the metadata for given ISSN through Crossref.
**Step 3** Fetch the ego network from COCI data for the DOIs of respective ISSN.
**Step 4** Merge these data to create scientific networks.
**Step 5** Apply the centrality analysis on these networks.

Python scripts are uploaded as Supplemental Files which can also be accessed on GitHub (*Butt & Faizi, 2020*). It gives replication and reuse of this study for other ISSN or bibliometric analyses for different network types. Details are provided below for the understanding of this study.

## Load citation network

Summary of the sub-steps to load citation network is shown in Fig. 3. Below we define the sub-steps to convert the COCI data to use in Python libraries for network processing. This step is computation and memory intensive but needs to be performed only once.

### Download COCI data

COCI is manually downloaded from *OpenCitations (2020)*. The 15 GB Zip file extracts to 98 GB set of files. Loading this data in-memory resulted in memory-overflow even when using 128 GB RAM. Therefore, in the next step, we remove the columns other than citing and cited. These two columns are used to create the article citation network.

### Convert COCI data to edge list

This step is needed to convert the COCI data to an edge list format. In this format, two nodes on each row signify an edge. This format is supported by SNAP (*Leskovec & Sosič, 2016*) for processing large-scale network data such as COCI. After this step, the edge list file is approx 35 GB. We convert the COCI from comma-separated-values (CSV) to space-separated-values having citing and cited columns. It is the only format supported by SNAP for bulk upload. Some formatting corrections are done for removing extra CR/LF and quotes. It hampers the loading process of SNAP. We have failed to load the same files with other libraries which are relatively more intuitive but not as powerful as SNAP (*Leskovec & Sosič, 2016*). However, we later discuss how this data can be used with other libraries. Details to save network in a format supported by most network processing libraries is provided in subsequent steps.

### Save COCI as binary

Loading 35 GB edge list in-memory using SNAP takes approx 5.5 h. Since the edge labels are DOI in the COCI data, therefore they are saved as strings. However, this slows down further processing so strings are converted to a hash file. There are two binary files generated when loading the COCI data in SNAP. First is DOIDirected.graph file which contains the directed citation network of COCI with integer node labels. Second is DOIMapping.hash which maps the integer node label to respective DOI. We save loaded graph as binary files for further computations. Loading binary file in-memory takes a few minutes, compared to hours for loading CSV data. Downside is that additional

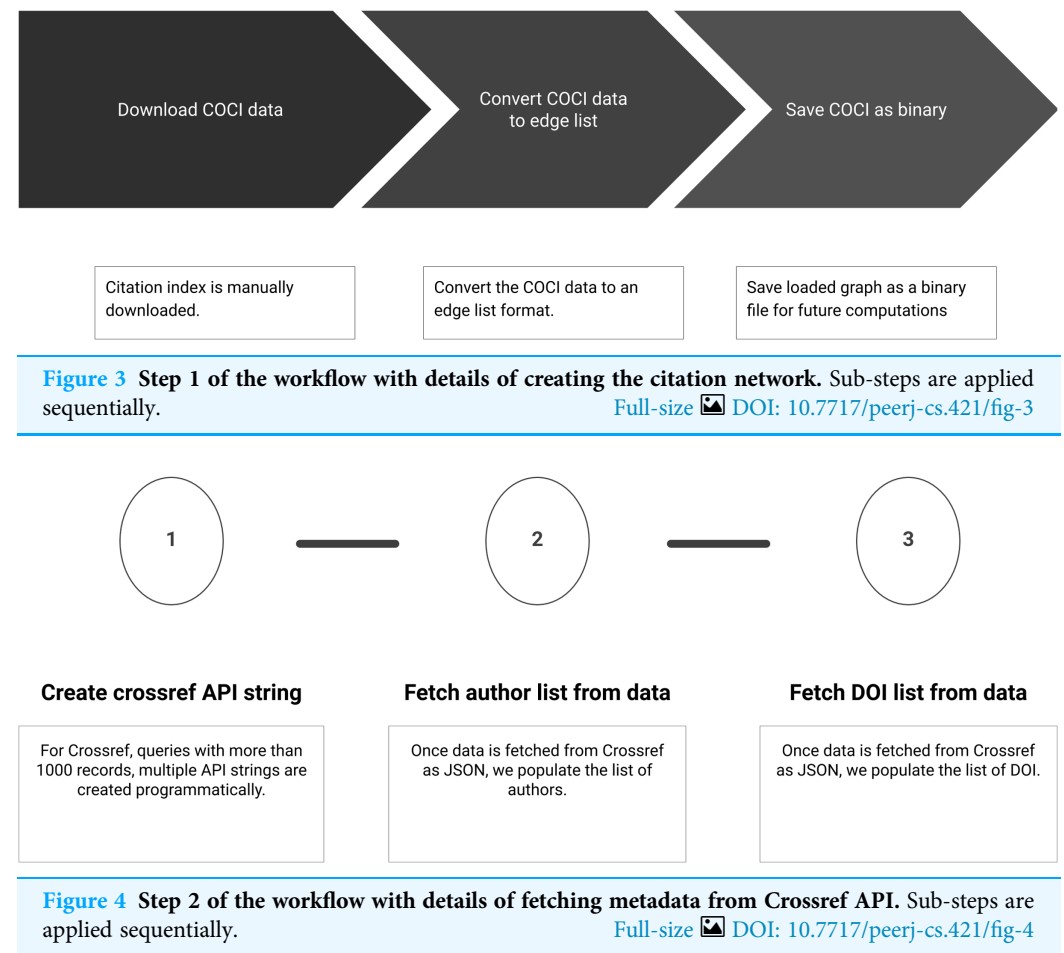

**Figure 3** **Step 1 of the workflow with details of creating the citation network.** Sub-steps are applied sequentially.                                     

**Figure 4** **Step 2 of the workflow with details of fetching metadata from Crossref API.** Sub-steps are applied sequentially.                                     

columns of COCI are currently not being utilised. DOIMapping.hash is simply a node list where node number is mapped to its label (DOI). DOIDirected.graph is an edge list on node number. Using numeric labels makes SNAP less intuitive but more powerful since computations are much faster when integer labels are used. The mapping to string labels is possible with the node list.

## Fetching Crossref metadata

Summary of the sub-steps to download Crossref metadata are shown in Fig. 4. Below, we define the sub-steps to fetch the citation metadata and converting it to list of authors and DOIs. These steps only give API string to fetch data for a single journal. However, it is possible to fetch data with other filters and details are available in Crossref documentation. The metadata downloaded via different filters is in a similar format which makes this script reusable for a variety of tasks.

### *Create Crossref API string*

Crossref limits a one time query to 1,000 records for a single ISSN. For queries with more than 1,000 records, multiple API strings are programmatically created. Metadata of SCIM is fetched via Crossref API which contains total 1,857 records. These records are fetched by two API requests and a combined JSON file is created.

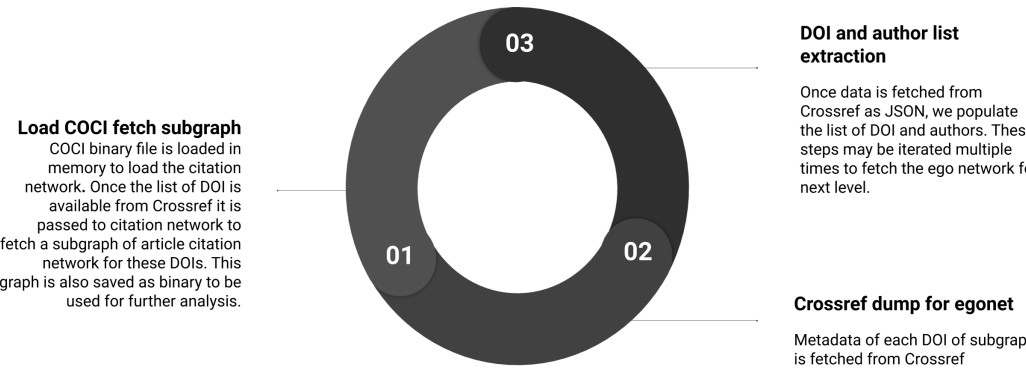

**Figure 5 Step 3 of the workflow with details of creating the ego network.** Sub-steps are applied sequentially, and may be iterated to create the next level of ego network.

### Fetch author(s) list from data

Once data are fetched from Crossref as JSON we populate the list of authors. We extract authors from the previous downloaded JSON. It is important to note that we do not apply any technique for author name disambiguation and rely on Crossref for correct author names. Although this is problematic for further analysis. Corrected data from a single source is much more efficient than using local methods of cleaning. A similar approach is used by MAG (*Wang et al., 2020*).

### Fetch DOI list from data

Once data are fetched from Crossref as JSON we populate the list of DOI. DOIs are extracted from the previously downloaded JSON. Although the purpose of fetching DOI is redundant, it's replica script is created to suggest that analysis with only given DOI list is also possible. So the previous two sub-steps can be ignored if analysing a specific journal is not needed. If the list of DOI is fetched from an external source then it can be easily incorporated in this workflow.

## Creating ego network

Summary of the sub-steps to create ego network are shown in Fig. 5. Below, we define the sub-steps to create ego network. This is an optional step. Iterating this step multiple times will grow the network as desired. This step is not used in the case study, however, with publicly accessible metadata it is easier to scale our approach. Further, this step justifies our approach of using SNAP over other network processing libraries. The process of creating the ego network is not only fast but intuitive to code due to a variety of functions available in the extensive library documentation. These functions make it easier to access the nodes in both directions of an edge. Also, the hash with integer labels makes the sub-graph computation faster than using string labels.

### Load COCI binary to fetch subgraph

After loading a binary file of COCI, a subset of the graph is fetched with nodes linked at one level apart. These nodes are either cited-by or cite the existing articles.

Processing a subgraph from 625M edges takes a few minutes on a Core i5 laptop with 16 GB RAM.

### Crossref dump for egonet

Crossref data are fetched for all DOIs of article ego network created in the previous step. First, all data gets downloaded and then processed to create the network. Depending on the size of the network, the number of ego levels, and connectivity bandwidth this process may continue from hours to days. Once a local copy of data is available this delay can be reduced. Since we do not have access to complete dump of Crossref, we could not identify that whether these same scripts can be reused. We assume that there would be few changes required to access the data locally.

### DOI and author list extraction

Processing of ego network for authors is similar to nodes of SCIM downloaded earlier. However, the connecting nodes fetched in subgraph above are added and their respective author details are downloaded.

## Creating scientific network(s)

Summary of the sub-steps to create scientific networks are shown in Fig. 6. Once all the data are pre-processed this step creates different types of network. We can also add bibliographic coupling and co-citation network within the list but they are ignored for two reasons. First, we did not find much evidence of centrality analysis on these networks. Secondly, the processing time for creating these networks for a very large citation network is relatively much longer than creating author collaboration or author citation network. These networks are created by making an edge list for authors who have collaborated or cited each other, respectively.

### Create article citation network

Once the list of DOI is available it is used to fetch subgraph of article citation network. Article citation network is fetched as a subgraph from COCI. Further, it saves the same graph as a binary file for further analysis. Also, the CSV file can be used with any graph processing library (such as NetworkX) while binary file can be read using SNAP.

### Create author collaboration network

Author collaboration is identified by the list of co-authors from JSON data fetched via Crossref. This refined data are further used in the case study in the subsequent section. It is important to note that the count of authors at this sub-step may vary from next sub-step of creating author citation network. The list of co-authors in Crossref is available as a list of names and do not include further metadata about these authors.

### Create author citation network

Using the subgraph of article citation network respective edges for authors are made to create author citation network. All co-authors are linked to use full counting method. In order to utilise partial counting method, this script needs to be modified. However, our workflow is not affected by the use of a partial or full counting method and hence we have

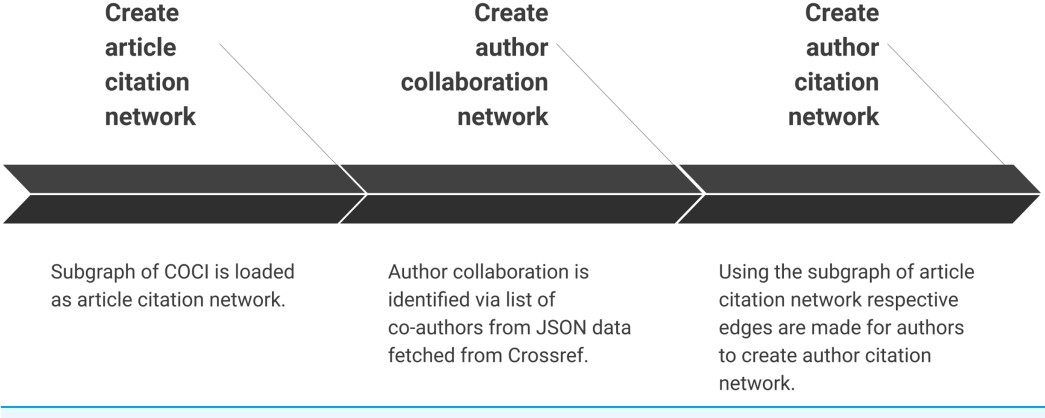

**Figure 6 Step 4 of the workflow with details of creating different scientific networks.** Sub-steps are applied sequentially.

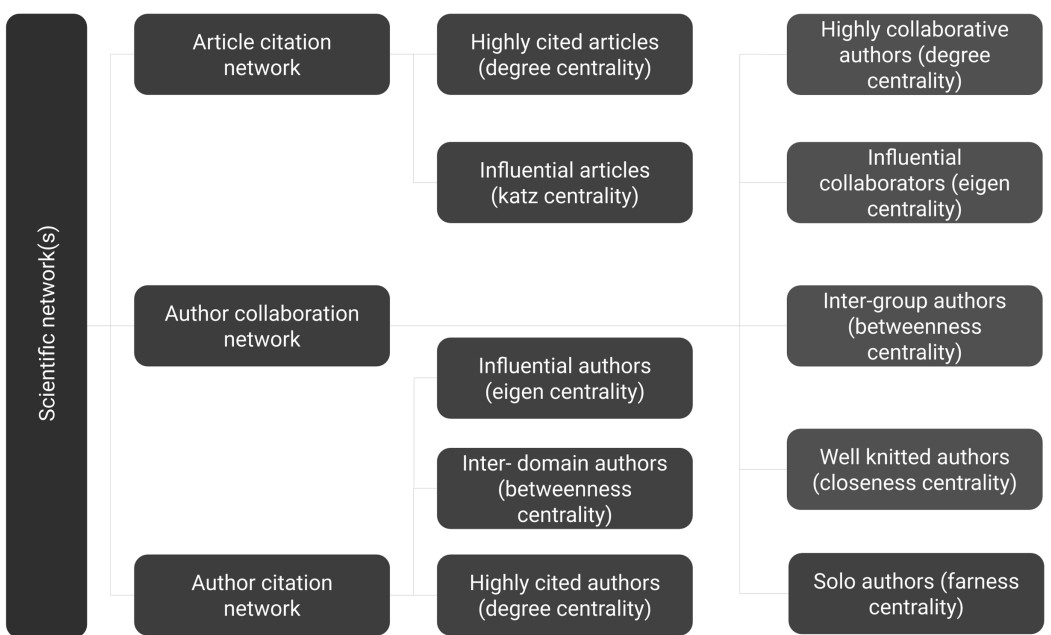

**Figure 7 Step 5 of the workflow with details of centrality measures that are applied on different scientific networks.** Sub-steps may be applied as required since there is no dependency within steps.

picked simpler one for brevity (*Glanzel, 2003*). This network shall supplement the analysis on the article citation network, as well as, the collaboration network that were constructed in the previous step.

## Centrality analysis

Summary of the sub-steps to apply centrality analysis are shown in Fig. 7. Below, we define the sub-steps to apply different centrality measures on the scientific networks. Based on numerous studies discussed in Table 1 it is evident that centrality measures are a popular way of identifying prominent authors. This is one of the common method employed in the bibliometric analysis and other methods of SNA can also be applied at this

**Table 2 Comparison of ranks by previous study (*Milojević, 2014*) and current study.** Table is divided into 4 sections for each centrality measure. The left column in each section showing the rank from *Milojević (2014)*, and the right column showing the rank calculated by our workflow.

| Collaborator name | Degree rank | | Betweenness rank | | Eigenvector rank | | Closeness rank | |
|---|---|---|---|---|---|---|---|---|
| | Prev | Curr | Prev | Curr | Prev | Curr | Prev | Curr |
| Glanzel, W | 1 | 1 | 1 | 1 | 1 | 2 | 1 | 1 |
| Rousseau, R | 2 | 2 | 3 | 2 | 16 | 1 | 5 | 2 |
| DeMoya-Anegon, F | 3 | 8 | 12 | 20 | 26 | 16 | 6 | 15 |
| Klingsporn, B | 4 | 3 | 21 | 16 | 89 | 121 | 174 | 144 |
| Ho, Ys | 5 | 5 | 22 | 125 | 2,096 | 3 | 613 | 575 |
| Thijs, B | 63 | 51 | 65 | 44 | 2 | 30 | 10 | 1,710 |
| Schubert, A | 36 | 48 | 38 | 28 | 3 | 18 | 27 | 24 |
| Debackere, K | 6 | 6 | 16 | 15 | 4 | 7 | 4 | 5 |
| Schlemmer, B | 670 | 832 | 382 | 962 | 5 | 808 | 33 | 37 |
| Meyer, M | 43 | 39 | 2 | 4 | 14 | 9 | 2 | 3 |
| Leydesdorff, L | 54 | 35 | 4 | 3 | 46 | 5 | 42 | 44 |
| Rafols, I | 1,058 | 387 | 5 | 23 | 83 | 239 | 45 | 49 |

step. Any tool or wrapper API may restrict the functionality at this point, however, this workflow can be extended to use any functions in existing network processing libraries.

### Applying centrality measures on article citation network

The article citation network is a Directed Acyclic Graph (DAG). Two measures are presented since most centrality analyses are not meaningful on DAG. Firstly, degree centrality gives highly cited articles. Secondly, influence definition in DAG is captured by Katz centrality.

### Applying centrality measures on author citation network

The author citation network has cyclic nature. Three measures are presented, namely highly cited authors (degree centrality), influential authors (eigenvector centrality), and authors working in multiple domains (betweenness centrality).

### Applying centrality measures on author collaboration network

The author collaboration network has cyclic nature and most centrality analyses are possible. Five measures are presented, namely highly collaborative authors (degree centrality), influential collaborators (eigenvector centrality), authors working in multiple groups (betweenness centrality), well-knitted authors (closeness centrality), and solo authors (farness centrality). Ranks captured here are presented in Table 2. This work was done manually by sorting individual lists on respective centrality scores and identifying their rank position.

## Batch execution

All python scripts can be executed through a sample batch file by modifying the ISSN and date range. This batch processing will also be useful for developing a front-end

tool, as well as modifying the sequence as per need. Details are available at *Butt & Faizi (2020)*.

## CASE STUDY USING SCIM

*Milojević (2014)* analysed collaboration network using WoS data of SCIM. Scores were calculated for degree, betweenness, eigenvector, and closeness centrality. The outcome of their analysis was provided in a table having authors that had top five ranks in either of the centrality scores. The respective rank of those authors for each centrality was also listed. To verify whether or not our workflow can capture a similar pattern we provide the results in a similar tabular form. For each of the centrality measure we provide the rank given in *Milojević (2014)* using WoS data. It is aligned with the rank obtained in our study using OpenCitatons data. We observe that the rank of authors for the degree, betweenness and closeness centrality is more or less similar. Further analysis is required to inquire the reason for the difference of eigenvector centrality ranks. Such an analysis is outside the scope of this study.

Further, we do not discuss the reasons for why a specific author has topped the list and what the centrality measure signifies, and the interested reader is referred to *Milojević (2014)*. Ranks in Table 2 were entered manually after execution of the workflow. Author names are placed in the same sequence as per the original study along with their respective ranks. Table 2 has four sections for the degree, betweenness, eigenvector and closeness centrality. Each section has two columns with the left column showing rank from *Milojević (2014)* and the right column shows the rank calculated for the same author using our workflow. It is pertinent to note that a very hands-on approach is given in *Milojević (2014)*. However, due to access restriction of WoS and its unaffordability for developing countries such useful analysis is only limited to researchers of specific institutes having WoS subscription (*Waltman & Larivière, 2020*). This highlights the importance of our workflow to provide access to any individual who can download the publicly available metadata.

## CONCLUSION AND FUTURE WORK

Scientific networks rely on completion of data (*Zingg, Nanumyan & Schweitzer, 2020*). Although the field has existed for more than 50 years (*Price, 1965*) but the limitations on data access have not helped to reach its true potential. We aim that with the availability of publicly available citation metadata (*Waltman & Larivière, 2020*) and a scripted workflow to access it (*Butt & Faizi, 2020*) a researcher from any field will be able to analyse the scientific networks. Its application can be vast, from identifying reviewers for a manuscript (based on article's references) to a graduate student finding a supervisor (through collaboration network). The time it takes to completely execute the workflow scripts is well under an hour, barring the two time intensive steps. First, saving citation index as a binary file which needs to be done only once. Second, downloading Crossref DOI files for individual nodes of ego-centered network can be optimised with a local copy. The workflow provides a means for fast and interactive analysis.

Since using graphical tools is easier than executing the scripts so a future application of this study is to create a front-end tool. A web-based portal is also under construction where the user may be able to select the date range along with other filters, and the system will initiate the scripts at the back-end. This way the researchers who are not familiar with programming can also benefit. It would enhance the capability and usefulness of this workflow. Techniques for author name disambiguation and partial counting have not been included. For effective analysis these need to be incorporated in future.

## ACKNOWLEDGEMENTS

The authors acknowledge lengthy discussions with Rauf Shams Malik and Azmi Umer that shaped this study, as well as, the efforts of students Sufyan Faizi, Adnan Anwar, Talha Aslam, Naveed Nizar, Sahil Shahbuddin and Anas ur Rehman in developing the Python scripts. Also acknowledge the invaluable comments of reviewers and editor that shaped this study.

### Funding

The authors received no funding for this work.

### Competing Interests

The authors declare that they have no competing interests.

### Author Contributions

- Bilal H. Butt conceived and designed the experiments, performed the experiments, analysed the data, performed the computation work, prepared figures and/or tables, authored or reviewed drafts of the paper, and approved the final draft.
- Muhammad Rafi conceived and designed the experiments, authored or reviewed drafts of the paper, and approved the final draft.
- Muhammad Sabih conceived and designed the experiments, authored or reviewed drafts of the paper, and approved the final draft.

### Data Availability

Python scripts are available in the Supplemental Files, GitHub (https://github.com/bilal-dsu/Guru-workflow/tree/v1.0), and Zenodo:

Bilal Hayat Butt, & Sufyan Faizi. (2020, November 11). Guru Workflow Scripts (Version v1.1). Zenodo. http://doi.org/10.5281/zenodo.4268321.

### Supplemental Information

Supplemental information for this article can be found online at http://dx.doi.org/10.7717/peerj-cs.421#supplemental-information.

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
