# Peer review of "A systematic metadata harvesting workflow for analysing scientific networks"

_PeerJ Computer Science, doi:10.7717/peerj-cs.421_

## Round 0.1 · original submission · Major Revisions

Dear authors,

Thanks for submitting your work at PeerJ Computer Science. Three independent experts have assessed your work, and you can find their reviews attached. There are several points of interest in your work, which are counterbalanced by significative issues. All agree that the article submitted is not acceptable for publication in the present form, and needs extensive rewriting before being ready for publication.

The main argument is about its main contribution. According to what you said, the contribution is twofold. On the one hand, to provide a workflow for retrieving open citation data and open bibliographic metadata for bibliometric/scientometric studies. On the other hand, you run an analysis using the data retrieved.

All the reviewers agree that the second part of the contribution, i.e. the analysis, should not be the focus of the work due to several flaws. However, all of them have praised the first part of the contribution, i.e. the workflow to run to download data.

Thus, my suggestion is to remove the part about the analysis and to solely focus on the workflow for downloading and processing the data.

Of course, there are issues that should also be addressed in the workflow part. Several of them are highlighted by the reviewers. In addition to those, I would add that making scripts available is not enough for claiming about the replicability of the workflow the scripts implement, but other resources must be made available as well. The whole workflow should be carefully described in the paper, with examples of use, a discussion of possible applications, some measures with respect to the quality of the networks created (see the issues about the author-based network below), etc.

I would also suggest the following additions:

- availability: all the code should be available in an online repository (e.g. GitHub)

- reusability: all the code should be released with appropriate open source licence to be reused by anyone

- workflow documentation: the repository must include appropriate documentation to enable a programmer to understand which data he/she needs and how to run all the Python scripts - better if they are defined as a proper protocol (e.g. see protocols.io) and cited in the article

- code documentation: all the code developed should be appropriately documented in order to understand how to use the various Python scripts and functions

- citation: the code developed should be citable, and thus it should be referred by using a DOI (e.g. via GitHub+Zenodo)

- general principles: since the article describes a software, software citation principles (https://peerj.com/articles/cs-86/) should be followed

In addition to that, there is an aspect of the workflow, which is not clear to me, and that concerns the author collaboration and citation networks. As far as I know, Crossref does not have a systematic way to disambiguate authors by using some identifiers. Indeed, Crossref has some ORCIDs specified in some entities, but the main part of them do not have any. Thus the problem is that, in the creation of the network that involves authors:
- how do you deal with the disambiguation of the authors?
- are two authors with the same name recognised as the same author?
- are two authors having the same family name and the same initial of the given name recognised as the same author?
- how you deal with homonymous authors?

All these issues should be taken into account when building such author-based networks in order to avoid mistakes.

Also, I strongly suggest to properly check the text, since the reviewers have highlighted several issues in using English to this respect.

Some typos:
- Open Citation data -> OpenCitations data

Your resubmission will be assessed again with a full reviewing session, since several modifications are necessary to prepare the revision of your work. Please let me know if you need more time for preparing the revision since the required modifications may take much effort to be completed.

Thanks again for having submitted to PeerJ Computer Science.
Have a nice day :-)

Silvio Peroni
https://orcid.org/0000-0003-0530-4305

Reviewer 1 ·

Basic reporting

The basic reporting is good, the very fact that the authors provide Pthon scripts shall be commended. The English is more or less OK, but the use or misuse of the capital letters is surprising. I show only a few errors:
"apply Network Analysis using open citation"
"Eigen centrality"?? You do not mean here Manfred Eigen, don't you?
"python script"

Some awkward constructions
"use case"
" instead of a software tool, we propose to use python libraries" But Python libraries ARE software tools
" identification of highly cited clusters of scientific clusters"

Experimental design

The research goal is clear, the methods are desrcibed with sufficient details

Validity of the findings

Dubious. Allthough the technique is valid, the validation protocol is problematic.

Additional comments

In this work B.H. Butt, M. Rafi, and M. Sabih demonstrate a software tool to download data from the Open Citation database and to process it. They demonstarte an example of such processing- measuring various kinds of centrality of the authors of a certain journal- Scientometrics.
While the protocol of downloading the whole Open Citation database shall be commended, the scientific benefits that the authors draw from the processing of this database are not clear enough. Hence, I do not suggest to publish this paper in its present form. However, , if the authors report only their downloading protocol and do not report data processing, such abridged paper is publishable.

Detailed comments.
The opening sentence of the paper "Identifying prominent authors (Gurus) of any field is one of the primary focus for researchers in that particular field" is blatantly wrong. A serious researcher working in some field shall know all prominent authors in his field without citation analysis of the corresponding databases. To identify prominent authors through citation databases- this is usually made by beginners or by the researchers in adjacent fields.

Some of the statements of the paper look like typos:
"Eigen centrality"- as if the author thinks that Eigen is a person.
"APS The American Phytopathological Society"- This is a very naive error. The abbreviation of the American Phytopathological Society is indeed APS, but in the field of informatics, APS means the American Physical Society and this is what the Ref. Singh and Jolad deals with.

"Data Source, primarily, is WoS or Scopus". Two of the authors come from the Department of Computer Science, why they are not familiar with CiteSeer?

Using their downloaded database, the authors choose to analyze the authors of the Scientometrics Journal. Thie goal is to identify the most prominent authors. However, one can't make such analysis basing on one journal- one needs to analyze the whole scientific field. There areseveral journals in this field: Scientometrics, Journal of Informetrics, Quantitative Science Studies, Journal of the American Society for Information Science and Technology. Prominent scientists in the the field of information science publish in these journals but due to rivalry between the journals, there is a certain association between some scientists and some journals. Moreover, european scientists tend to publish in the european journals. Hence, by analyzing citation network of one journal, it is impossible to find all prominent figures in the field- one shall analyze all journals atogether.

Reviewer 2 ·

Basic reporting

The authors go in great detail to explain what they have done, yet the article is lacking on several points. The language is not always used appropriately, several figures are low-resolution, and most definitions are given without reference to the related papers nor formally (e.g., for centralities).

Experimental design

The research questions are well-defined, yet unfortunately they do not focus on novel contributions.

RQ1 explores whether network centrality measures can be used to detect popular authors, or "gurus", which is something that has been extensively explored in previous work.

RQ2 is, instead, more novel in that it attempt to use an open citation index, COCI, and compare its results with WoS. Yet, this study has also been recently performed (in much greater detail): https://arxiv.org/abs/2004.14329

Validity of the findings

I do not have much to add in terms of findings, as the main issue with the paper is in its lack of novelty. I would, nevertheless, suggest to the authors to share their code (per se, a great thing to do) using online persistent repositories such as Zenodo.

·

Basic reporting

See my general comments.

Experimental design

See my general comments.

Validity of the findings

See my general comments.

Additional comments

The contribution of this paper is in providing a set of Python scripts for performing scientometric network analyses in a reproducible manner based on open data sources. The paper does not aim to make a substantive contribution by providing new scientometric insights. I value the work presented in the paper. However, the authors need to be more clear about the contribution and the scope of their paper. For instance, I believe the following sentence needs to be removed from the abstract: “We have shown that centrality analysis is a useful measure for identifying prominent authors.” The paper does not show this. Likewise, the authors claim to answer the following research question: “Is it possible to identify prominent authors (Gurus) of any field, by applying Centrality measures on Scientific Networks?” I don’t believe the paper answers this question (except by summarizing some earlier literature, but this is not an original contribution). It therefore seems to me that this research question needs to be removed from the paper. The discussion section at the end of the paper also needs to be revised accordingly.

“Does the coverage of CrossRef (for Open Access publishers), hamper the Network Analysis as compared to WoS, or it can be replicated?”: This research question needs to be rephrased. Crossref provides data not only for open access publishers but also for subscription-based publishers, and for both types of publishers most data is openly available. Also, since the empirical analysis presented in the paper focuses on a single journal (Scientometrics), the paper offers only a partial answer to the question whether WoS-based analyses can be replicated using Crossref data.

The comparative analysis section is hard to understand. According to the authors, Milojevic (2014) “have fetched the data at least 5 years earlier than us, therefore, the total citation count is different”. This is difficult to understand. It is not clear to me how exactly the authors collected their data, and in particular for which time period data was collected. I don’t understand why the authors didn’t organize their data collection in such a way that it is as similar as possible to the data collection performed by Milojevic (2014). Having data sets that are as similar as possible is essential for a meaningful comparative analysis. If there are basic differences in the time periods covered by two data sets, I don’t see the value of performing a comparative analysis.

Ego networks play a central role in the paper, but the paper doesn’t provide a proper explanation of what an ego network is. A one-sentence explanation is provided on p. 2, but this is not sufficient. The authors should provide a more extensive discussion of ego networks and their relevance in scientometric analyses.

Also, while most of the paper is about analyzing ego networks, these networks are not considered at all in the empirical part of the paper. It would be very helpful if the authors could add a section to their paper in which they give a practical example of an analysis of an ego network.

“the results of a case study based on WoS data is reproduced to confirm that accessing metadata of publishers, which submit metadata to CrossRef as Open Access (such as Springer) does not hamper analysis as compared to WoS. However, the same would not be true for Publishers whose data is not yet available with CrossRef (such as Elsevier).”: The information in these sentences is not entirely correct. Both Springer and Elsevier make basic metadata such as titles and author lists of publications openly available in Crossref. They also both make reference lists of publications available in Crossref. The only difference between the two publishers is that Springer makes reference lists openly available, while Elsevier keeps reference lists closed.

Most readers won’t understand the discussion on partial vs. full counting on p. 9. This requires some additional explanation.

Throughout the paper ‘CrossRef’ should be written as ‘Crossref’.

The authors may be interested in a special issue on bibliographic data sources published in the first issue of Quantitative Science Studies: https://www.mitpressjournals.org/toc/qss/1/1. This special issue offers a lot of information on data sources such as WoS, Crossref, and OpenCitations.

---

## Round 0.2 · Minor Revisions

Dear authors,

Thanks for submitting your revised work at PeerJ Computer Science. The same three independent experts who assessed your initial submission were able to review your revision again, and their reviews are attached. All of them praised the work and the extensive rewriting you did in the article. However, there are still some issues to address before this article is acceptable for publication in PeerJ Computer Science.

Please read carefully the reviewers' comments and address all of them in the new revision of your work. Please let me know if you need more time for preparing the revision.

Thanks again for having submitted to PeerJ Computer Science.
Have a nice day :-)

Silvio Peroni
https://orcid.org/0000-0003-0530-4305

Reviewer 1 ·

Basic reporting

Although the authors reworked their submission according to suggestions of the reviewers, I find too many inconsistencies to recommend its publication in its present form.

While the review of previous studies and description of the workflow are good, the formulation of the research goals and the case study are problematic.

Indeed, the abstract formulates the research goal as developing a methodology to construct a multinetwork from the same dataset. Namely, citation network overlayed with authorship network and collaboration network. This is an ambitious program and the authors seem to be able to show the workflow, algorithm how they perform this task. Such algorithms are known and the authors shall be commended for their thorough description of how these algorithms work together, in tandem or pipeline.

The authors present the example of such workflow, the case study. They downloaded the authors writing to Scientometrics, calculated different centralities associated with them, and compared their measurements to previous study (Milojevic, 2014). Table 1 show dramatic difference between their ranking of prominent authors and those of Milojevic, In other words, the authors shoot themselves in the foot, since discrepancies in Table 1 invalidate their algorithm.

My feeling is that either description of some details of their algorithm is missing, or they do not explain in which aspect their measurements are different from those of Milojevic.

Experimental design

good.

Validity of the findings

Problematic. Probably, some details of the algorithm are missing.

Reviewer 2 ·

Basic reporting

See below.

Experimental design

See below.

Validity of the findings

See below.

Additional comments

Dear authors,

thank you for a thoughtful and extensive revision of your article. I believe its goals and scope are now clear, as well as its contribution. The article is also now well-embedded into previous literature, and the publication of the code on GitHub is crucial.

While I remain skeptical about the actual scientific contribution of this work, which I consider somewhat narrow, I believe that the authors have substantially improved on their previous submission and, if the editor considers their work of interest to PeerJ readers, I now support acceptance.

·

Basic reporting

See my general comments.

Experimental design

See my general comments.

Validity of the findings

See my general comments.

Additional comments

I would like to thank the authors for the improvements they have made to their paper. Before I can recommend this paper for publication, there are some further improvements that I consider to be necessary.

The introduction of the paper, in particular the first paragraph of the introduction, needs to provide a better explanation of what the paper is about. The introduction is largely focused on discussing the problem of identifying ‘gurus’. This gives the incorrect impression that the paper may provide in-depth analyses of different approaches to identifying gurus. The introduction does not make sufficiently clear that the paper is almost entirely about providing a standardized workflow for identifying gurus. This needs to be made much more clear in the introduction.

The subsection ‘Research question’ in the section ‘Preliminaries’ can best be removed. If the authors want to specify an explicit research question, it is best to do so in the introduction.

The distinction between the section ‘Discussion’ and the section ‘Conclusion and future work’ is not very clear. My recommendation is to merge the two sections in a single section.

“Any study on co-authorship may not necessarily have all the citation links.”: I don’t understand this sentence. To study co-authorship, there is no need to have information about citation links. A study of co-authorship requires data on co-authorship. It doesn’t require citation data.

“the same would not be true for publishers whose metadata are not yet public although available with Crossref (such as Elsevier).”: As I pointed out in my previous review report, this is incorrect. Elsevier does make metadata openly available in Crossref. The only exception is citation data, which Elsevier currently doesn’t make openly available in Crossref. (By the way, this will change soon. See https://www.elsevier.com/connect/advancing-responsible-research-assessment.)

---

## Round 0.3 · Minor Revisions

Dear authors,

Thanks again for your revision. I think that the current form of the article is fine for being accepted for publication in PeerJ Computer Science, pending a few additional suggestions that can be addressed directly on the camera-ready version of the paper – and that I will personally check before publishing the article.

1. In the new part of the introduction, you refer to a "common researcher". However, it is not clear at all to what that "common" actually refers to, in particular in comparison to what? What is an "uncommon researcher" then? I think it is important to clarify explicitly what kind of users you are going to help with the implementation of the workflow you are proposing. Is your work of any help to researchers (in what? Scientometrics?) with no expertise in programming? Or, does it address issues that researchers with expertise in programming but no expertise in Scientometrics may have in retrieving and analysing these data? Or, again, is that done for helping data scientists? etc. Thus, you need to clarify in the introduction which specific users (i.e. kinds of researchers) are you going to help with your computational workflow.

2. When you refer to open source software, such as NetworkX and SNAP, please mention it as it is. In particular, "open access" should not be used with software, "open source" should.

3. The license specified in the release on GitHub (https://github.com/bilal-dsu/Guru-workflow/tree/v1.0) is CC0, which does not apply to software applications. Please, choose an appropriate license to use for releasing the software - e.g. see the list at https://opensource.org/licenses.

4. In the GitHub readme, there should be explicitly stated how to call every single Python script developed, since some of them actually take in input parameters, and they are not defined in the text. Suppose to be one of your users. By reading the text it is clear what each script does, but it is not clear how to run it properly.

5. You say that your focus is "to have a systematic workflow to identify prominent authors (gurus) using publicly available metadata for citations". However, as far as I understood, the point is slightly different. Indeed, the workflow you devised is for collecting data and calculating metrics that then **can be used** to identify gurus, but the identification of gurus is not the focus of the present work. Thus, please avoid stressing too much on this aspect. This is also reflected by the fact you are presenting a case study to show one possible use of the workflow and not full comparative research on gurus identification. Please, relax a bit your claim about what this article is about.

6. In your answers to reviewers, you say that "Research Question is merged with the introduction section". However, I cannot see any research question stated there. Actually, it seems that you have totally removed it from the article. While this can be fine, after looking at your answer I expected to find it in the introduction of the revision. Is the current form (i.e. no research questions) correct, or did you miss to add it in the introduction?

7. In the conclusion you state that "for case study, some manual work was also done to sort and format the results, however, it can also be scripted in future as it does not hamper the workflow and can be performed as a standalone". I thought that the workflow was complete, and thus it could retrieve all the data you need to perform the case study. But here you say that "manual work" (outside of the workflow, I presume) was needed to address the case study. I think you should clarify this passage, otherwise, it seems that, in the end, the case study is not reproducible despite the workflow you implemented.

8. I would suggest revising the sentence in the abstract, i.e. "Any study on co-authorship may not need all the citation links, however, for a holistic view citation links may also be needed", in a way which is more compliant with the reviewer's comment in the previous round of reviews. In particular, it may be stated that while studies on co-authorship do not need, usually, information about the citation network, having citation links would enable a more complete and holistic view of the possible relations among authors.

9. Please, check again the English of the whole article, since there are several typos, mistakes and long and ambiguous sentences that should be rewritten to drastically improve the readability of the text.

10. Typos:
- libraries of Python --> Python libraries
- line 91: the citation should be to (Heibi et al., 2019), not to (Peroni and Shotton, 2020)
- It seems that the DOI of the Guru script is not correct
- References: please, check the consistency of all the references, and add DOI URLs when possible


Thanks again for your revision.
Have a nice day :-)

Silvio Peroni
https://orcid.org/0000-0003-0530-4305

---

## Round 0.4 · accepted · Accept

Dear authors,

Thanks for your revision and for having addressed all my points. I believe that now your contribution is appropriate and can be accepted in PeerJ Computer Science. As a minor issue, please check carefully again the language in the paper since I have found a bunch of typos in the text.

---

## Author Rebuttal · Round 0.4

Editor,
PeerJ CS                                                              21 Jan 2021

Thank you for your encouraging comments. We have revised language of the manuscript. Abstract and introduction have been restructured to conform to the comments. Also, GitHub readme has been updated.

Please see point-wise response below.

Thanks,

Bilal, Rafi and Sabih

###############################################################################################
Thanks again for your revision. I think that the current form of the article is fine for being accepted for publication in PeerJ Computer Science, pending a few additional suggestions that can be addressed directly on the camera-ready version of the paper – and that I will personally check before publishing the article.

1. In the new part of the introduction, you refer to a "common researcher". However, it is not clear at all to what that "common" actually refers to, in particular in comparison to what? What is an "uncommon researcher" then? I think it is important to clarify explicitly what kind of users you are going to help with the implementation of the workflow you are proposing. Is your work of any help to researchers (in what? Scientometrics?) with no expertise in programming? Or, does it address issues that researchers with expertise in programming but no expertise in Scientometrics may have in retrieving and analysing these data? Or, again, is that done for helping data scientists? etc. Thus, you need to clarify in the introduction which specific users (i.e. kinds of researchers) are you going to help with your computational workflow.
**>> response: Our primary focus is to target bibliometricians, however, researchers looking to inspect their own field may utilize the workflow as well. A basic understanding of programming is required.**
**Having said that, this workflow may be utilized by a front-end developer to create a graphical tool. Therefore, we do not want to explicitly mention a user base. We developed a prototype dashboard and willing to enhance it in future. Just sharing a sample image.**

[Figure]

2. When you refer to open source software, such as NetworkX and SNAP, please mention it as it is. In particular, "open access" should not be used with software, "open source" should.
**>> response: Correction done.**

3. The license specified in the release on GitHub (https://github.com/bilal-dsu/Guru-workflow/tree/v1.0) is CC0, which does not apply to software applications. Please, choose an appropriate license to use for releasing the software - e.g. see the list at https://opensource.org/licenses.
**>> response: Updated to MIT License.**

4. In the GitHub readme, there should be explicitly stated how to call every single Python script developed, since some of them actually take in input parameters, and they are not defined in the text. Suppose to be one of your users. By reading the text it is clear what each script does, but it is not clear how to run it properly.
**>> response: A separate batch file is provided. Details have also been added on readme.**

5. You say that your focus is "to have a systematic workflow to identify prominent authors (gurus) using publicly available metadata for citations". However, as far as I understood, the point is slightly different. Indeed, the workflow you devised is for collecting data and calculating metrics that then **can be used** to identify gurus, but the identification of gurus is not the focus of the present work. Thus, please avoid stressing too much on this aspect. This is also reflected by the fact you are presenting a case study to show one possible use of the workflow and not full comparative research on gurus identification. Please, relax a bit your claim about what this article is about.
**>> response: Along with restructuring abstract and introduction, we have also modified the title to reflect this understanding.**

6. In your answers to reviewers, you say that "Research Question is merged with the introduction section". However, I cannot see any research question stated there. Actually, it seems that you have totally removed it from the article. While this can be fine, after looking at your answer I expected to find it in the introduction of the revision. Is the current form (i.e. no research questions) correct, or did you miss to add it in the introduction?
**>> response: Current form is correct.**

7. In the conclusion you state that "for case study, some manual work was also done to sort and format the results, however, it can also be scripted in future as it does not hamper the workflow and can be performed as a standalone". I thought that the workflow was complete, and thus it could retrieve all the data you need to perform the case study. But here you say that "manual work" (outside of the workflow, I presume) was needed to address the case study. I think you should clarify this passage, otherwise, it seems that, in the end, the case study is not reproducible despite the workflow you implemented.
**>> response: Manual work incorporated was to enter the ranks in Table 2. Workflow retrieves the complete data.**

8. I would suggest revising the sentence in the abstract, i.e. "Any study on co-authorship may not need all the citation links, however, for a holistic view citation links may also be needed", in a way which is more compliant with the reviewer's comment in the previous round of reviews. In particular, it may be stated that while studies on co-authorship do not need, usually, information about the citation network, having citation links would enable a more complete and holistic view of the possible relations among authors.

**>> response: We have placed the suggested sentence in the introduction and have also made a relevant citation to it.**

9. Please, check again the English of the whole article, since there are several typos, mistakes and long and ambiguous sentences that should be rewritten to drastically improve the readability of the text.

**>> response: Text has been rewritten.**

10. Typos:
- libraries of Python --> Python libraries
- line 91: the citation should be to (Heibi et al., 2019), not to (Peroni and Shotton, 2020)
- It seems that the DOI of the Guru script is not correct
- References: please, check the consistency of all the references, and add DOI URLs when possible

**>> response: Incorporated.**